# Comparison of visual performance between monofocal and rotationally asymmetric refractive intraocular lenses

**Masaki Miyoshi**[1☯], **Hirotaka Tanabe**[1*☯], **Tomohiro Shojo**[1], **Tomofusa Yamauchi**[1], **Kosuke Takase**[1], **Hitoshi Tabuchi**[1,2]

**1** Department of Ophthalmology, Tsukazaki Hospital, Himeji, Japan, **2** Department of Technology and Design Thinking for Medicine, Hiroshima University Graduate School of Biomedical and Health Sciences, Hiroshima, Japan

☯ These authors contributed equally to this work.
* tennsyoudragon@icloud.com

## Abstract

We compared the visual performance of a monofocal intraocular lens (IOL) (ZCB00) and a rotationally asymmetric refractive IOL with +1.5 diopters near addition (Lentis Comfort LS-313 MF15) by investigating the ten-week postoperative parameters of both eyes of cataract patients who underwent bilateral implantation with one of the two IOLs within three months between 2011 and 2019. A total of 1352 eyes of 676 patients were enrolled; the ZCB00 group comprised 904 eyes of 452 patients (72.3±6.8 years; females/males, 268/184), and the LS-313 MF15 group included 448 eyes of 224 patients (73.6±7.0 years; females/males, 125/99). Comparisons were made with a linear mixed-effects model, strictly adjusting for sex, age, subjective refraction spherical equivalent, subjective refraction cylinder, corneal astigmatism, axial length, corneal higher-order aberrations, and pupil diameter to ensure statistical validity. The corrected distance visual acuity was significantly better, and most of the higher-order aberrations were significantly smaller in the ZCB00 group (p<0.00068, Wald test). Contrast sensitivity with glare (6.3 degrees) and spectacle independence (near) were significantly better in the LS-313 MF15 group (p<0.00068, Wald test).

## Introduction

Numerous studies have demonstrated that multifocal intraoperative lenses (IOLs) provide good distance to intermediate or near vision [1–4]. However, they are also known to cause complications such as halo, glare, and reduced contrast, which have been suggested to significantly reduce patients' postoperative satisfaction [4–9].

In Japan, a rotationally asymmetric refractive intraocular lens with +1.5 diopters near addition (Lentis Comfort LS-313 MF15 [Teleon Surgical BV, Spankeren, Netherlands]) has been available as an insured medical treatment since April 2019. It is a

**Data availability statement:** All relevant data are within the manuscript and its Supporting Information files.

**Funding:** The author(s) received no specific funding for this work.

**Competing interests:** The authors have declared that no competing interests exist.

one-piece, multifocal IOL composed of hydrophilic, transparent acrylic material, with a segmental aspheric optical section in one half for distance and in the other half that provides +1.5 D near addition. The lens has an optical loss of only 5%, resulting in good postoperative contrast sensitivity [10–13].

Another lens, a monofocal IOL (ZCB00 [Johnson & Johnson Surgical Vision, Inc., Santa Ana, CA, USA]), is made of hydrophobic transparent acrylic material. It is a one-piece, single-focal IOL with a 6.0 mm optical section that has a flat aspheric front surface and negative spherical aberration, designed to minimize spherical aberration across the eye and improve contrast after cataract surgery [14–16].

In Japan, the ZCB00 is one of the most commonly used monofocal IOLs, whereas the LS-313 MF15, which features a rotationally asymmetric design and a +1.5 diopter near addition, was the first nonmonofocal IOL to be approved by insurance. Given their substantial impact on Japanese society, we conducted a retrospective study to compare the visual performance of these two types of IOLs. The study was based on the data accumulated over a decade of practice at a single eye institute.

## Materials and methods

### Design

Retrospective Comparative Case Series

### Setting

Ophthalmology, Tsukazaki Hospital, Japan

### Patients

We reviewed the data of a consecutive case series of cataract patients who underwent bilateral implantation of Tecnis monofocal IOLs (ZCB00) or rotationally asymmetric refractive IOLs with +1.5 diopters near addition (Lentis Comfort LS-313 MF15) within 3 months from December 13, 2010, to December 28, 2019, in the same way as in our previous study [17,18]. Participants, including outpatients with or without a doctor's referral, were recruited for enrollment in a consecutive case series study. This study was conducted in a way that prevented any potential self-selection bias that could have influenced the results. To be eligible for this study, participants had to not meet certain exclusion criteria, such as a history of ocular diseases that could potentially affect visual function, |subjective equivalent (SE)| greater than 2.00 diopters, |subjective refraction cylinder (CYL)| greater than 3.00 diopters, and |corneal astigmatism (keratometric cylinder)| greater than 3.00 diopters 10 weeks after surgery.

### Preoperative examination

Before surgery, all patients underwent a comprehensive eye examination identical to that in our previous studies [17–20]. The examination included assessments of the curvature of the cornea, corneal astigmatism, axial length, refractive status, ocular aberrations, pupil size, visual acuity at different distances, and contrast sensitivity

with and without glare. Additionally, the anterior segment of the eye was evaluated using slit-lamp microscopy, tonometry, and indirect fundoscopy. The quality of vision was assessed using the Japanese version of the NEI VFQ-25 [21], administered by experienced nurses or technicians who interacted with the patients face-to-face. The use of spectacles was also evaluated by asking the patients how frequently they used them for distance, intermediate, and near vision; the possible responses were "never," "sometimes," or "always."

Both uncorrected distance visual acuity (UDVA) and corrected distance visual acuity (CDVA) were assessed at 5.0 m, while, uncorrected intermediate visual acuity (UIVA) and corrected intermediate visual acuity (CIVA) were both measured at 0.5 m. Similarly, at a distance of 0.3 m, uncorrected near visual acuity (UNVA) and corrected near visual acuity (CNVA) were measured. In order to measure visual acuity, a decimal visual acuity chart was used; the decimal values obtained were then converted to the logarithm of the minimum angle of resolution (logMAR). The corneal curvature radius, corneal astigmatism, and objective refractive status were measured using a KR-8900 autorefractor keratometer manufactured by Topcon in Tokyo, Japan. The axial length was measured using an IOL Master biometer manufactured by Carl Zeiss in Oberkochen, Germany and an AL-3000 biometer manufactured by TOMEY in Nagoya, Japan. Contrast sensitivity and contrast sensitivity under glare were measured using a CGT-1000 contrast glare tester manufactured by Takagi Seiko. Lastly, the pupil diameter and ocular aberrations were measured using a KR-1W wavefront analyzer manufactured by Topcon in Tokyo, Japan. It is worth noting that all measurements were taken by experienced technicians.

### IOLs and surgical technique

After being informed of the benefits and drawbacks of each type of lens, the patients opted for implantation of either monofocal or rotationally asymmetric refractive IOLs, as described in our previous studies [17–20]. Patients in the monofocal group were implanted with Tecnis monofocal IOLs (ZCB00) bilaterally, while those in the rotationally asymmetric refractive group received the rotationally asymmetric refractive IOLs with +1.5 diopters near addition (Lentis Comfort LS-313 MF15) bilaterally. The ZCB00 lens features a 6.0-mm acrylic optics design that is hydrophobic and clear. It also features a modified prolate and aspherical anterior surface, which helps to reduce spherical aberrations and enhance contrast sensitivity under mesopic conditions following cataract surgery [14–16]. The Lentis Comfort LS-313 MF15 has a clear, hydrophilic, transparent acrylic 6.0-mm optics design [13] and a segmental aspheric optical section for the upper half for distance and for the lower half of + 1.5 D near addition, with a smooth transition zone between the far and near portions of the optics, reducing optical loss. Experienced technicians were responsible for acquiring all measurements. Twenty experienced cataract surgeons operated on the patients using a standard technique that involved sutureless microincision phacoemulsification and the same protocol. The surgical procedures included creating a scleral or corneal incision measuring 1.8 to 2.8 mm in length, performing 5 mm continuous capsulorhexis, phacoemulsification cataract extraction, and implanting the IOL with an injector. Topical anesthesia was used during the surgeries.

### Postoperative examination

A protocol identical to the preoperative assessment was followed to evaluate patients 10 weeks after the surgery.

### Statistical analyses

The required number of eyes for the study was calculated based on an alpha value of 0.00068 and a power of 0.80. We also took into consideration a VA standard deviation of 0.10 logMAR units and a minimum detectable difference of 1 line of VA (0.1 logMAR), which was derived from our previous study [17]. According to the results of the calculation, we needed to include 39 eyes per group. The study included 904 eyes of 452 patients in the monofocal group and 448 eyes of 224 patients in the rotationally asymmetric refractive group, which was sufficient to meet the required sample size.

As in our previous studies [17–20], we compared the two groups—those who received monofocal IOLs and those who received rotationally asymmetric refractive IOLs—in terms of various postoperative measures, which were assessed 10 weeks after surgery for both eyes. The postoperative parameters included visual acuity (uncorrected/corrected, distance/intermediate/near), contrast sensitivity (with/without glare), and higher-order aberrations (ocular/internal, scaled to a pupil size of 4 mm/6 mm), which were assessed via mixed-effects linear regression. The VFQ-25 score was evaluated using a linear regression model or logistic regression. Spectacle dependence (distance/intermediate/near) was analyzed using cumulative logistic regression. Both groups were adjusted for age, sex, axial length, subjective refraction spherical equivalent, subjective refraction cylinder, corneal astigmatism, corneal higher-order aberrations, and pupil diameter. For regression analyses related to the VFQ-25 score and spectacle dependence, the data were divided into two parts (left-eye data and right-eye data), and the regression model was applied to each part. Certain VFQ-25 scores (Peripheral_Vision, Color_Vision, Driving_Daytime, Driving_Nighttime, and Driving_Adverse_Conditions) were treated as binary data due to their discrete natures. Patients were divided into two groups based on their scores (75 or lower and above 75), and a logistic regression model was applied to each group. Finally, the results of the left- and right-eye analyses were combined using the inverse variance method; the corrected values were calculated for the left- and right-eye datasets before computing the average values.

The Wald test was utilized in the regression analysis to assess the significance of differences in postoperative parameters between the two groups. After Bonferroni's correction, the significance level was established at 0.00068. Correlation analysis was executed separately for the monofocal and rotationally asymmetric refractive groups to examine the relationship among postoperative parameters. For each group, a heatmap indicating Pearson's correlation coefficients was produced. The two-sided t test was used to determine if the correlation coefficients were significantly different from zero, with a level of significance set to 0.00002 after applying Bonferroni's correction.

The commercially available software program R (version 3.6.1; R Core Team, 2019, Vienna, Austria) [22] was used to conduct the statistical analyses.

### Ethics statement

The tenets of the Declaration of Helsinki were followed in this study, which was approved by the Ethics Committee of Tsukazaki Hospital. Each subject provided written informed consent in accordance with applicable guidelines and regulations for all research conducted. This study, "Performance comparison among different intraocular lenses in cataract surgery," was registered under number UMIN000035630. We accessed the data for research purposes from February 27, 2019, after registration of the research. We made the information anonymous after data collection.

## Results

### Patient characteristics

S1 Table shows the pre/postoperative visual parameters and patient demographics. A total of 1352 eyes from 676 patients were included in the study. The monofocal group consisted of 904 eyes of 452 patients (72.3±6.8 years; 268 [59.3%] females and 184 [40.7%] males), while the rotationally asymmetric refractive group included 448 eyes of 224 patients (73.6±7.0 years; 125 [55.8%] females and 99 [44.2%] males).

### Comparison of postoperative parameters between patients implanted with the monofocal IOLs (ZCB00) and those implanted with the rotationally asymmetric refractive IOLs with +1.5 diopters near addition (Lentis Comfort LS-313 MF15)

S2 Table displays the postoperative parameters for both eyes of the monofocal and rotationally asymmetric refractive groups and the results of multiple regression analysis following adjustment by the explanatory variables in Table 1 and

**Table 1. Parameters of the Monofocal and Rotationally Asymmetric Refractive Groups Used to Adjust the Linear Regression Model: Age, Sex, Axial Length (at the Time of Surgery), Subjective Refraction Spherical Equivalent (SE), Subjective Refraction Cylinder (CYL), Corneal Astigmatism (Keratometric Cylinder), Corneal Higher-Order Aberrations (Astigmatism, Total HOA, Third, Fourth, Trefoil, Coma, Tetrafoil, 2ndAstig, Spherical, Scaled to a Pupil Size of 4 mm/6 mm), and Pupil Diameter (10 Weeks after Surgery).**

| Variable | ZCB00 | LS-313 | P value (Wald test) |
|---|---|---|---|
| Categorical variable, N (%) | | | |
| Sex | | | |
| F/M | 268 (59.3)/184 (40.7) | 125 (55.8)/99 (44.2) | 4.339E-01 |
| Continuous variables, N, Mean±SD | | | |
| Age | 452, 72.3±6.82 | 224, 73.6±7.03 | 3.868E-04* |
| SE | 762, 0.11±0.474 | 368, -0.22±0.562 | 3.771E-21* |
| CYL | 640, -0.96±0.535 | 301, -0.91±0.514 | 1.870E-01 |
| Corneal Astigmatism | 754, -0.78±0.461 | 420, -0.81±0.496 | 4.113E-01 |
| Axial Length | 900, 23.54±1.283 | 447, 23.73±1.113 | 1.905E-05* |
| WF_4_post_C | | | |
| Astigmatism | 581, -1.00±0.631 | 323, -0.954±0.601 | 3.897E-01 |
| Total HOA | 581, 0.23±0.127 | 323, 0.23±0.105 | 8.082E-01 |
| Third | 581, 0.20±0.123 | 323, 0.20±0.101 | 5.772E-01 |
| Fourth | 581, 0.11±0.059 | 323, 0.010±0.050 | 7.814E-03* |
| Trefoil | 581, 0.16±0.102 | 323, 0.15±0.088 | 9.345E-01 |
| Coma | 581, 0.11±0.092 | 323, 0.12±0.079 | 1.573E-01 |
| Tetrafoil | 581, 0.07±0.047 | 323, 0.06±0.037 | 5.620E-04* |
| 2ndAstig | 581, 0.05±0.042 | 323, 0.05±0.031 | 5.425E-01 |
| Spherical | 581, 0.05±0.045 | 323, 0.04±0.049 | 1.976E-01 |
| WF_6_post_C | | | |
| Astigmatism | 500, -0.74±1.021 | 256, -0.69±0.468 | 6.999E-01 |
| Total HOA | 500, 0.70±1.285 | 256, 0.62±0.350 | 2.479E-01 |
| Third | 500, 0.47±0.894 | 256, 0.44±0.262 | 4.845E-01 |
| Fourth | 500, 0.44±0.719 | 256, 0.37±0.194 | 3.917E-05* |
| Trefoil | 500, 0.35±0.636 | 256, 0.32±0.231 | 6.303E-01 |
| Coma | 500, 0.29±0.646 | 256, 0.26±0.190 | 5.771E-01 |
| Tetrafoil | 500, 0.21±0.466 | 256, 0.16±0.169 | 8.633E-07* |
| 2ndAstig | 500, 0.13±0.457 | 256, 0.11±0.113 | 3.552E-01 |
| Spherical | 500, 0.32±0.345 | 256, 0.29±0.118 | 4.904E-02* |
| Pupil Diameter_post | 592, 4.19±0.811 | 334, 3.90±0.797 | 4.976E-08* |

In this table, categorial variables are presented as the counts and frequencies, whereas continuous variables are presented with the number of samples (excluding those with missing data) and the mean±standard deviation. Fisher's two-sided exact test was utilized to compare the categorical data between the monofocal and rotationally asymmetric refractive IOL groups. The two-sided Mann-Whitney U test was used to compare the numerical data between the monofocal and rotationally asymmetric refractive IOL groups.

* p<0.05

SE: subjective refraction spherical equivalent; CYL: subjective refraction cylinder; WF_4_post_C: wavefront_4_post_corneal; HOA: higher-order aberration.

according to the same protocol used in our previous study [17–20]. The results were obtained 10 weeks after surgery. Corrected distance visual acuity and the higher-order aberrations (ocular/internal, scaled to a pupil size of 4 mm/6 mm) WF_4_post_O_TotalHOA, _Third, _Fourth, _Trefoil, _Coma, _Tetrafoil, _Spherical, WF_4_post_I_TotalHOA, _Third, _Fourth, _Trefoil, _Coma, _Tetrafoil, _Spherical, WF_6_post_O_Spherical, and WF_6_post_I_Spherical were significantly better in the monofocal group (p<0.00068, Wald test) (Table 2 and Fig 1), and the uncorrected distance visual acuity, the

**Table 2. Parameters that Were Significantly Different at p<0.00068 or p<0.05 between the Monofocal and Rotationally Asymmetric Refractive Groups 10 Weeks Postsurgery for Both Eyes.**

| Response | After adjusting | | Coefficients (95% CI) | P value (Wald test) |
|---|---|---|---|---|
| | ZCB00 | LS-313 | | |
| UDVA | -0.01±0.11 | 0.04±0.10 | 0.05 (0.02, 0.08) | 1.717E-03* |
| CDVA | -0.13±0.06 | -0.09±0.06 | 0.03 (0.01, 0.05) | 5.245E-04** |
| UNVA | 0.56±0.15 | 0.46±0.17 | -0.07 (-0.12, 0.02) | 1.235E-02* |
| CNVA | 0.02±0.04 | 0.05±0.09 | 0.02 (0.01, 0.04) | 6.164E-03* |
| WF_4_post_O | | | | |
| TotalHOA | 0.20±0.07 | 0.29±0.07 | 0.09 (0.07, 0.11) | 2.266E-16** |
| Third | 0.18±0.07 | 0.26±0.06 | 0.08 (0.06, 0.10) | 7.242E-14** |
| Fourth | 0.09±0.03 | 0.13±0.03 | 0.04 (0.03, 0.05) | 5.710E-16** |
| Trefoil | 0.13±0.06 | 0.18±0.05 | 0.05 (0.03, 0.07) | 1.036E-09** |
| Coma | 0.11±0.04 | 0.17±0.05 | 0.05 (0.04, 0.07) | 7.696E-10** |
| Tetrafoil | 0.06±0.02 | 0.07±0.02 | 0.02 (0.01, 0.03) | 4.533E-06** |
| Spherical | 0.02±0.03 | 0.08±0.03 | 0.06 (0.05, 0.07) | 2.070E-39** |
| WF_4_post_I | | | | |
| Astigmatism | -0.62±0.13 | -0.70±0.13 | -0.09 (-0.15, 0.02) | 1.500E-02* |
| TotalHOA | 0.14±0.06 | 0.27±0.06 | 0.12 (0.11, 0.14) | 3.901E-32** |
| Third | 0.11±0.05 | 0.25±0.05 | 0.13 (0.11, 0.14) | 8.759E-38** |
| Fourth | 0.08±0.03 | 0.10±0.03 | 0.02 (0.01, 0.03) | 4.584E-05** |
| Trefoil | 0.07±0.04 | 0.17±0.04 | 0.09 (0.08, 0.10) | 2.609E-43** |
| Coma | 0.08±0.04 | 0.17±0.03 | 0.09 (0.07, 0.10) | 1.406E-21** |
| Tetrafoil | 0.05±0.03 | 0.07±0.02 | 0.03 (0.02, 0.03) | 6.084E-09** |
| Spherical | -0.02±0.03 | 0.04±0.03 | 0.06 (0.05, 0.07) | 2.070E-39** |
| WF_6_post_O | | | | |
| TotalHOA | 1.61±0.95 | 2.28±1.02 | 0.47 (0.14, 0.80) | 5.238E-03* |
| Third | 0.86±0.45 | 1.21±0.50 | 0.27 (0.04, 0.50) | 2.089E-02* |
| Trefoil | 0.58±0.35 | 0.90±0.38 | 0.28 (0.08, 0.49) | 7.838E-03* |
| Spherical | 0.47±0.36 | 0.83±0.36 | 0.35 (0.18, 0.51) | 3.110E-05** |
| WF_6_post_I | | | | |
| TotalHOA | 1.46±0.98 | 2.13±1.04 | 0.47 (0.13, 0.80) | 6.498E-03* |
| Third | 0.75±0.50 | 1.13±0.51 | 0.30 (0.07, 0.53) | 1.114E-02* |
| Trefoil | 0.47±0.37 | 0.82±0.39 | 0.32 (0.11, 0.52) | 2.783E-03* |
| Spherical | 0.16±0.35 | 0.54±0.34 | 0.35 (0.18, 0.51) | 3.110E-05** |
| Contrast Sensitivity | | | | |
| C_6.3 | 0.03±0.01 | 0.02±0.01 | -0.01 (-0.01, 0.00) | 5.697E-03* |
| Contrast Sensitivity with Glare | | | | |
| G_6.3 | 0.04±0.02 | 0.03±0.01 | -0.02 (-0.02, 0.01) | 7.840E-06** |
| G_4.0 | 0.05±0.03 | 0.04±0.03 | -0.02 (-0.03, 0.01) | 3.709E-03* |
| G_1.6 | 0.10±0.05 | 0.08±0.04 | -0.02 (-0.04, 0.01) | 8.235E-03* |
| G_1.0 | 0.18±0.07 | 0.16±0.07 | -0.03 (-0.05, 0.00) | 2.821E-02* |
| G_0.7 | 0.34±0.08 | 0.30±0.08 | -0.04 (-0.07, 0.01) | 1.257E-02* |
| VFQ-25 | | | | |
| General_Health | 57.49±6.48 | 63.16±5.30 | 5.91 (2.37, 9.44) | 1.049E-03* |
| Driving_Adverse_ Conditions (Low/High) | 24/23 | 31/7 | -0.69 (-1.37, 0.01) | 4.831E-02* |
| Spectacle Dependence | | | | |

*(Continued)*

**Table 2.** (Continued)

| Response | After adjusting | | Coefficients (95% CI) | P value (Wald test) |
|---|---|---|---|---|
| | ZCB00 | LS-313 | | |
| Intermediate | 74/0/9 | 55/0/1 | -0.78 (-1.55, 0.01) | 4.829E-02* |
| Near | 3/0/80 | 31/0/25 | -1.67 (-2.18, 1.16) | 1.535E-10** |

Multiple regression was used to adjust each parameter with explanatory variables listed in Table 1. The mean and standard deviation for each numerical parameter or the counts for each categorical parameter (spectacle dependence: never/sometimes/always) are provided for each response variable. The regression coefficient, its 95% confidence interval, and the p value (Wald test) are also included.

\* $p < 0.05$, \*\* $p < 0.00068$

UDVA: uncorrected distance visual acuity; CDVA: corrected distance visual acuity; UNVA: uncorrected near visual acuity; CNVA: corrected near visual acuity; C: contrast sensitivity; G: contrast sensitivity under glare; WF_4_post_O: wavefront_4_post_ocular; WF_4_post_I: wavefront_4_post_internal; HOA: higher-order aberration.

corrected near visual acuity, the higher-order aberrations WF_4_post_I_Astigmatism, WF_6_post_O_TotalHOA, Third, Trefoil, WF_6_post_I_TotalHOA, Third, and Trefoil, and the VFQ-25 score for driving in adverse conditions were slightly but significantly better in the monofocal group ($p < 0.05$, Wald test) (Table 2).

Contrast sensitivity with glare (6.3 degrees) and spectacle independence (near) were significantly better in the rotationally asymmetric refractive group ($p < 0.00068$, Wald test) (Table 2 and Figs 1 and 2), and uncorrected near visual acuity, contrast sensitivity (6.3 degrees), contrast sensitivity with glare (4.0/1.6/1.0/0.7 degrees), the VFQ-25 score for general health, and spectacle independence (intermediate) were slightly but significantly better in the rotationally asymmetric refractive group ($p < 0.05$, Wald test) (Table 2 and Figs 2 and 3).

### Correlation among postoperative parameters of the monofocal IOL (ZCB00) and the rotationally asymmetric refractive IOL with +1.5 diopters near addition (Lentis Comfort LS-313 MF15) groups

S3 Table and S4 Table present the correlation coefficients (A) and p values for the correlation analyses (B) between all possible combinations of postoperative parameters for the monofocal and rotationally asymmetric refractive groups, respectively. These parameters have been adjusted through multiple regression using the explanatory variables illustrated in Table 1, following the same methodology as in our previous studies [17–20].

## Discussion

In our previous retrospective study comparing the visual performance of patients who received a monofocal intraocular lens (ZCB00) or a multifocal intraocular lens (ZMB00), based on data accumulated over a decade of practice in a single ophthalmic institute [17], superior contrast sensitivity and higher night driving scores were observed in the ZCB00 group. As in the previous studies [17–20], the CGT-1000 instrument was used to evaluate contrast sensitivity; this device determines the contrast threshold with an automated strategy for 6 spatial frequencies [23,24]. In the present study, the contrast sensitivity with glare was significantly superior in the rotationally asymmetric refractive group at 6.3 degrees. Additionally, the contrast sensitivity without glare was also better in the rotationally asymmetric refractive group at 6.3 degrees (Figs. 1 and 2). Aspheric IOLs with negative spherical aberrations are known to help mitigate ocular spherical aberrations and increase contrast sensitivity [14–16]. Although only the ZCB00 has a negative spherical aberration to compensate for corneal positive spherical aberrations, both the ZCB00 and the Lentis Comfort are aspheric IOLs. Previous reports have shown that a rotationally asymmetric refractive IOL can provide contrast sensitivity comparable to that of a monofocal IOL [13]. In the present study, the contrast sensitivity of the rotationally asymmetric refractive IOL was comparable to or partially exceeded that of the monofocal IOL. The Lentis Comfort IOL is a multifocal intraocular lens designed to reduce optical loss. It features a smooth transition between the near and far portions of the lens, with only a mild difference in focus

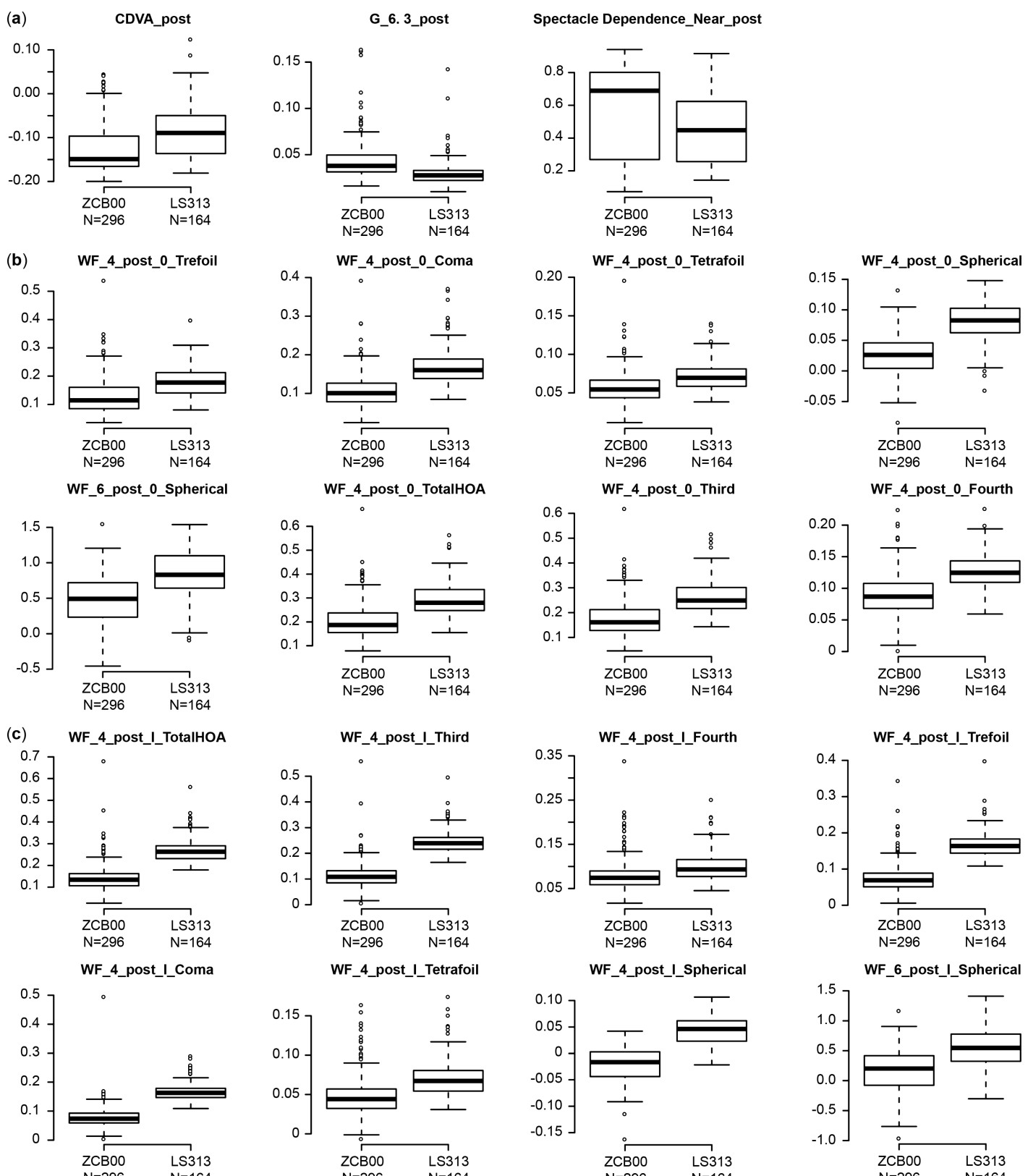

**Fig 1. Significantly Different Parameters between the ZCB00 and LS-313 MF15 Groups 10-weeks after Bilateral Eye Surgery.** The band within the box represents the median. To highlight any potential outliers, the upper whisker is set as the maximum or third quartile+ 1.5×IQR, and the lower

whisker indicates the minimum or first quartile-1.5×IQR. Each parameter was adjusted via multiple linear regression with the explanatory variables listed in Table 1. A two-sided Wald test was used to assess the significance of differences between the two groups, and the level of significance was set to 0.00068 using Bonferroni's correction. Non-WF (a), WF_4/6_post O (b), and WF_4/6_post I (c).

between each area; this design contributes to a mere 5% reduction in contrast sensitivity. Furthermore, unlike the ZCB00 IOL, the Lentis Comfort IOL is hydrophilic, which has been reported to increase the Abbe number [25]; indeed, although the Abbe number of the ZCB00 IOL is very high (55), that of the Lentis Comfort IOL is even greater (57) according to the manufacturer's reports. The Abbe number, or Vd-number, measures a transparent material's dispersion, indicating how the refractive index changes with wavelength. High Abbe numbers indicate low dispersion, which is crucial in the design of an IOLs; a lower Abbe number suggests greater chromatic aberration, resulting in decreased retinal image quality. The refractive indices of ocular media vary with wavelength, resulting in different focal points for colors and leading to chromatic aberrations and blurred images. The cornea and lens are the primary anatomical structures that contribute to these aberrations, with the crystalline lens responsible for approximately 28.5% of the total [26]. While typically manageable for patients with phakic eyes, longitudinal chromatic aberration can impact visual function in pseudophakic eyes equipped with low Abbe number IOLs, making the Abbe number of the IOL a key factor in contrast sensitivity differences [27,28]. For all the reasons listed above, the Lentis Comfort IOL seemed to have slightly but significantly better contrast sensitivity (6.3 degrees), contrast sensitivity with glare (4.0, 1.6, 1.0, 0.7 degrees), and significantly better contrast sensitivity with glare (6.3 degrees). In other words, there were no differences in most degrees of contrast sensitivity between the Lentis Comfort and ZCB00 IOLs in this study, despite the sufficient number of patients and rigorous statistical analysis involving a linear mixed-effects model with binocular data adjusted for age, sex, subjective refraction cylinder (CYL), subjective refraction spherical equivalent (SE), corneal astigmatism, axial length, pupil diameter, and corneal higher-order aberrations to ensure accurate results. Nevertheless, other confounding factors could explain the results.

In this study, a significantly better spectacle independence rate was observed in the rotationally asymmetric refractive group at short distances, while a slightly but significantly better spectacle independence rate was observed at intermediate distances. This is consistent with previous data.

The National Eye Institute Visual Function Questionnaire (NEI VFQ-25) assesses the self-reported vision-targeted health status of individuals with chronic eye diseases [21,29–34]. In our previous study, better uncorrected intermediate/near visual acuity and spectacle independence were observed in patients implanted with multifocal IOLs, while superior contrast sensitivity and higher scores for night driving were demonstrated in patients implanted with monofocal IOLs of the same material [17]. In this study, the general health score of the VFQ-25 was slightly but significantly higher in the rotationally asymmetric refractive group, whereas that for driving in adverse conditions was slightly but significantly higher in the monofocal group (Figs 1 and 3). These results imply that the Lentis Comfort group had higher general health figures due to better near and intermediate spectacle independence, while the ZCB00 group had better driving in adverse conditions scores because their lenses were designed to fully correct for ocular spherical aberrations.

A heatmap was generated for the monofocal IOL group, displaying correlation coefficients between all possible variable combinations. The coefficients were adjusted using multiple regression and the explanatory variables from Table 1. The heatmap in the monofocal IOL group revealed a strong correlation between the VFQ-25 total items, Distance vision, and General health scores with UDVA and CDVA (S1 Fig). However, the heatmap in the rotationally asymmetric refractive IOL group revealed low correlation coefficients between UDVA/CDVA and the VFQ-25 items score (S2 Fig), which suggests there could be other factors related to the VFQ-25 related health status in the rotationally asymmetric refractive group or that the results of the statistical analyses require more patients for detecting statistical significance. In the monofocal IOL group, there was a correlation between UDVA/CDVA and contrast sensitivity at 4.0–0.7 degrees and contrast sensitivity with glare at 1.6–0.7 degrees. On the other hand, in the rotationally asymmetric refractive group, there was a correlation

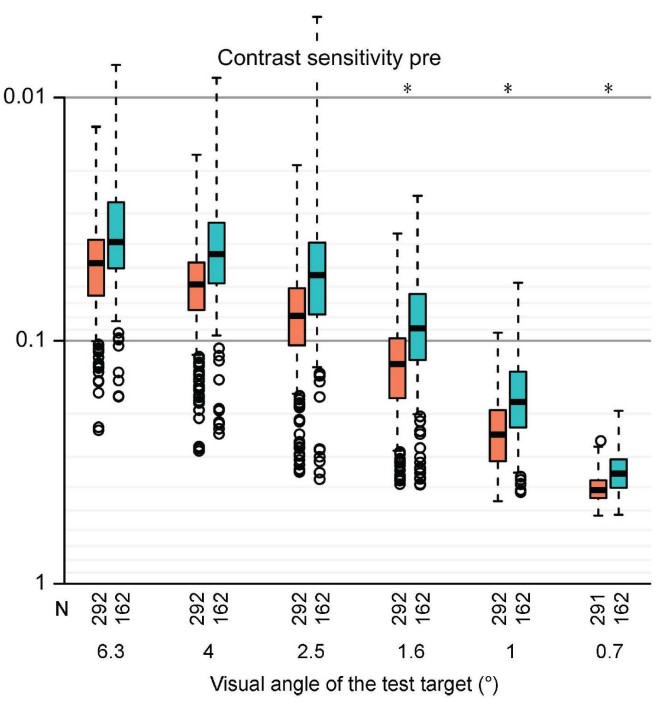

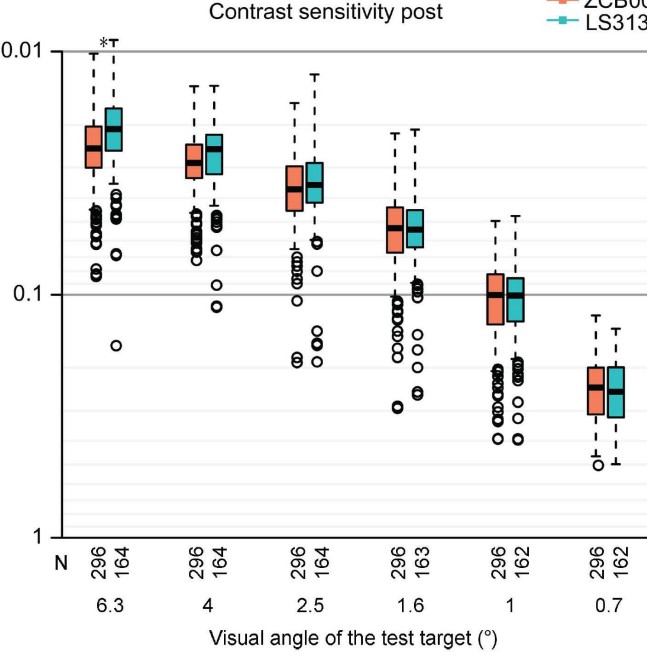

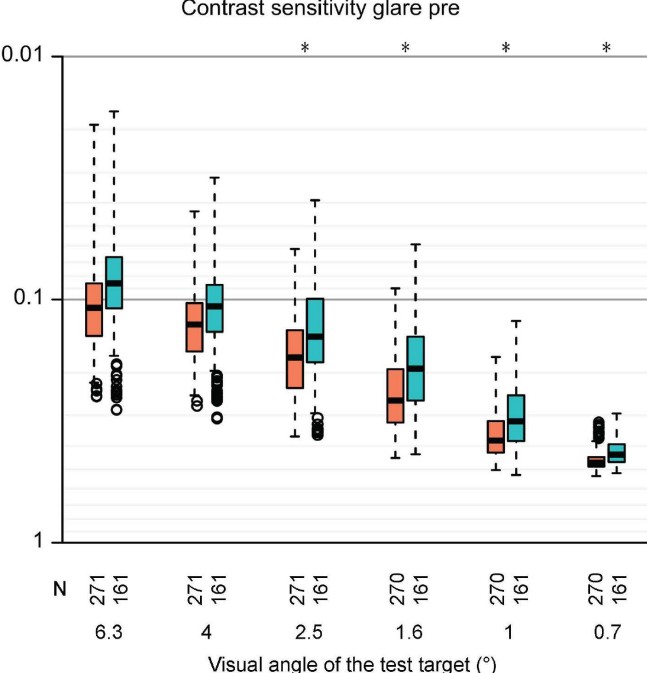

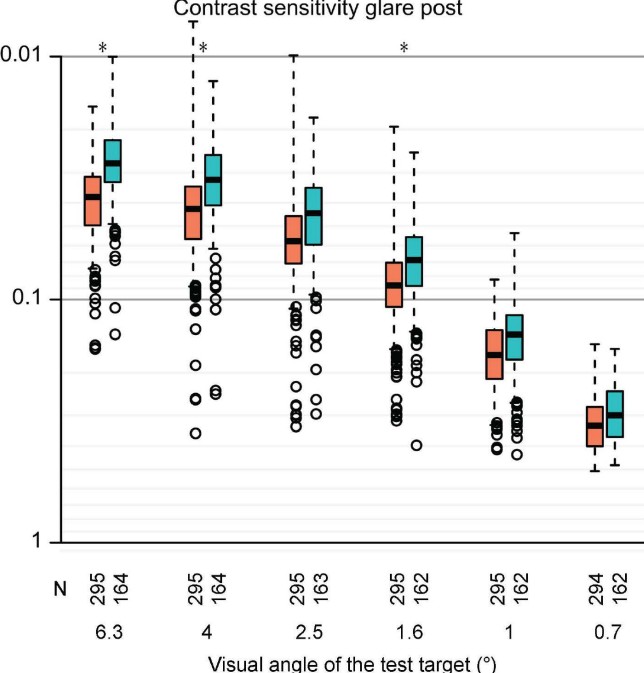

**Fig 2. Contrast Sensitivity with/without Glare in the ZCB00/LS-313 MF15 Groups before/10 Weeks after Bilateral Eye Surgery.** Box-and-whisker plots are used to represent the data, where the bottom of the box indicates the first quartile and the top of the box indicates the third quartile. The median is represented by the band inside the box. To identify potential outliers, the upper whisker is set to the maximum or third quartile + 1.5 × IQR, and the lower whisker indicates the minimum or first quartile -1.5 × IQR. All parameters were adjusted by multiple linear regression with the explanatory variables listed in Table 1. The two-sided Wald test was conducted to assess the significance of differences between the two groups, and a significance level of 0.0083 was set after Bonferroni's correction.

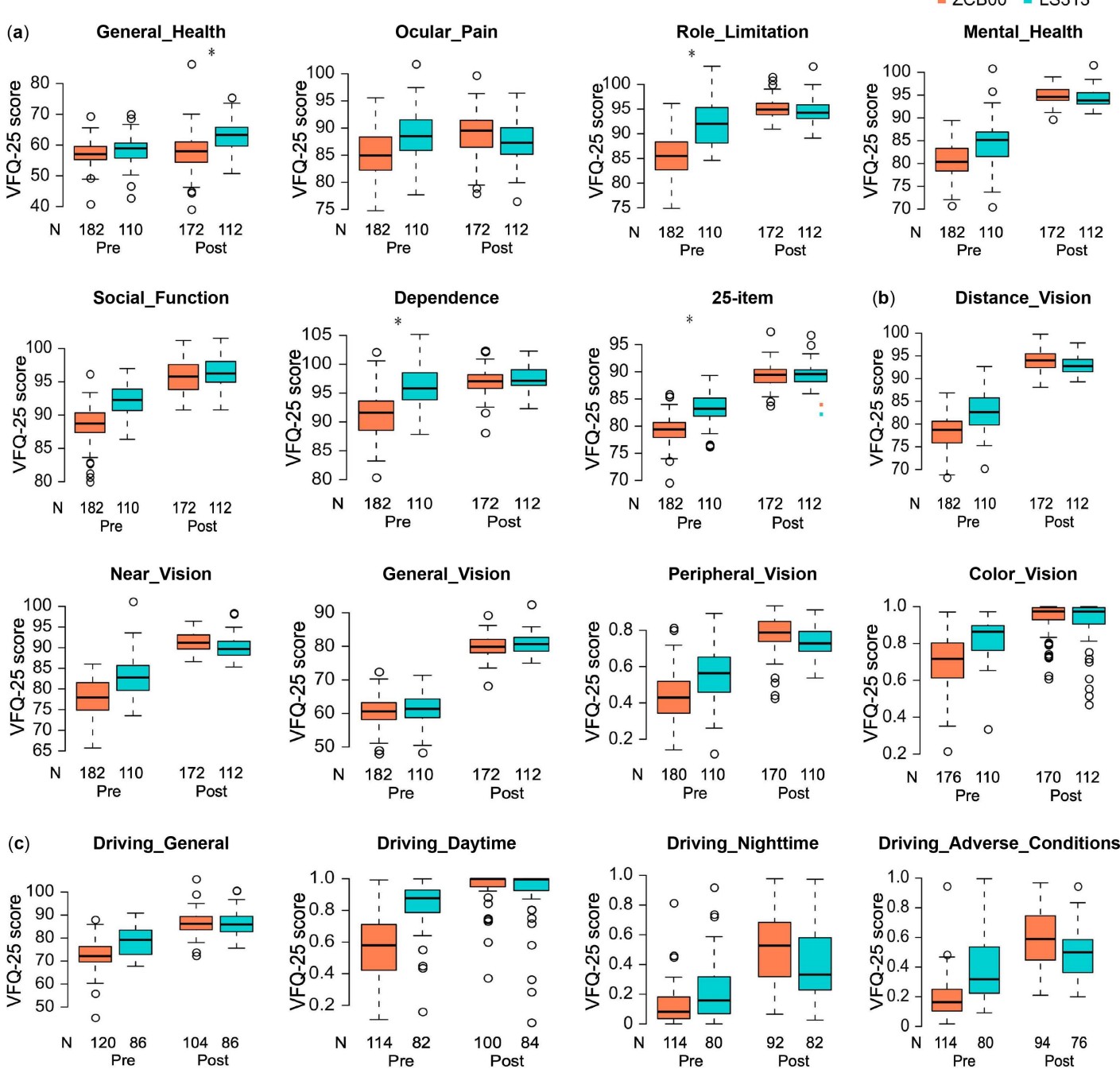

**Fig 3. VFQ-25 Scores in the ZCB00/LS-313 MF15 Groups before/10 Weeks after Bilateral Eye Surgery.** The box-and-whisker plots show the first quartile at the bottom of the box and the third quartile at the top. The median is represented by the band inside the box. To highlight suspected outliers, the upper whisker is set as either the maximum or third quartile + 1.5 × IQR, and the lower whisker indicates the minimum or first quartile - 1.5 × IQR. The explanatory variables in Table 1 were used to adjust each parameter via multiple linear regression. Differences between the two groups were evaluated using the two-sided Wald test, with a significance level of 0.003125 after Bonferroni correction. In this figure, an asterisk (*) denotes a significant difference between two groups at a p < 0.003125. Nonvision/nondriving **(a)**, vision **(b)**, and driving **(c)**.

between CDVA and contrast sensitivity at 4.0, 1.0, and 0.7 degrees and contrast sensitivity with glare at 6.3 degrees and 2.5–0.7 degrees. Interestingly, there were significant positive correlations between the higher-order aberrations (ocular/internal, scaled to a pupil size of 6 mm) and distance spectacle dependency in the rotationally asymmetric refractive IOL group (S2 Fig), which suggests that a certain pupil size is needed to use the portion of the plate for near vision, and larger higher-order aberrations at a 6 mm pupil setting, which could be partially due to a mixture of the different focus powers, might have led to the increase in the distance spectacle dependency in this group.

It should be noted that one limitation of this study is that intermediate visual acuity was only measured at 50 cm and near visual acuity at 30 cm, using the same method as in our previous study [17–20]. While there has been historical controversy regarding the distance at which the examination is conducted [35], it is ideal to measure visual acuity at various distances to accurately assess the performance of the lenses. In particular, a distance within arm's reach is practical and has been shown to be an important factor, especially considering that Japanese individuals tend to have a shorter average height and arm length than European or American individuals. Therefore, we measured intermediate visual acuity at 50 cm, which we assumed would be within reach for most patients in the study.

Furthermore, this was a retrospective study, and it is possible that there were differences in the socioeconomic statuses of the patients between the groups. However, this study was conducted in a single center with a large patient group, and we adhered to the same procedure outlined in our earlier research [17–20] to maximize the validity of the results. We confirmed that all surgeons carefully adhered to the same protocols, ensuring thorough standardization of the procedures. As a result, we believe that variation among surgeons is minimal. Therefore, we did not include the surgeon as a random effect in the mixed-effects model used in this analysis when accounting for any potential biases in performance. The presence of multiple surgeons, each with a sufficient number of patients, resulted in a robust sample for our study. A total of 1,352 eyes from 676 patients met the selection criteria for this study, which is significantly larger than the required sample size determined from the power calculations on the basis of the logMAR effect size derived from previous studies. Since the objective of this study was also to conduct an exploratory analysis of trends for other observable factors, we chose to employ the entire population for our analysis. Another reason for this decision was to avoid any potential biases that could arise from selectively sampling the population under analysis. Before surgery, written informed consent was obtained from all the patients. We evaluated the same set of pre- and postoperative parameters, including the VFQ-25 score, 10 weeks after the last surgery in cataract patients who underwent ZCB00 or LS-313 MF15 implantation in both the right and left eyes within 3 months of each other. We strictly adjusted for age, sex, axial length, subjective refraction SE, subjective refraction CYL, corneal astigmatism (keratometric cylinder), corneal higher-order aberrations, and pupil diameter. We also utilized a linear mixed model to account for bias in our analysis due to the mixture of items evaluated either in both eyes or in each eye separately. Additionally, we corrected for multiple observations for each eye per patient. We made sure that every patient who underwent lens implantation was randomly and independently selected, and that all endpoints were measured, despite the retrospective nature of the study. It is generally assumed in statistical analysis that random assignment does not introduce bias in the results, even if there are differences in the number of patients, as is the case with the 1:n allocation used in clinical trials.

In conclusion, we compared the visual performance of a monofocal IOL and a rotationally asymmetric refractive IOL. The monofocal group exhibited better corrected distance visual acuity and smaller higher-order aberrations, while the rotationally asymmetric refractive group demonstrated superior contrast sensitivity with glare (6.3 degrees) and near spectacle independence. In terms of visual parameters, each IOL group displayed different characteristics at a high level of performance.

## Supporting information

**S1 Table. Patient Demographics and Pre- and Postoperative Visual Parameters.** Categorical data are shown as the count and frequency, and Fisher's exact test (two-sided) was used for comparison between the monofocal and rotationally asymmetric refractive IOL groups. For numerical data, the mean and standard deviation are shown, and the Mann–Whitney U test (two-sided) was used for comparison between the monofocal and rotationally asymmetric refractive IOL groups.

(XLSX)

**S2 Table. Results of Multiple Regression Analysis of all Postoperative Parameters of the Monofocal (ZCB00) and Rotationally Asymmetric Refractive (LS313 MF15) IOL Groups 10 Weeks after Surgery for Both Eyes.** For numerical parameters, multiple mixed linear regression or multiple linear regression was applied, while cumulative logistic regression was applied to spectacle dependency parameters. In the multiple linear regression or cumulative logistic regression, the variables in Table 1 were used as the explanatory variables. The regression coefficients, their 95% confidence intervals, and the p values (Wald test) are shown in (a) for each response variable. The original and corrected values (i.e., before and after adjusting with multiple linear regression) of the mean and standard deviation for each numerical parameter and the counts for each categorical parameter (spectacle dependency: never/sometimes/always), regression coefficient, 95% confidence interval, and p value (Wald test) are shown in (**b**).
(XLSX)

**S3 Table.** Pearson's correlation coefficients (a) and p values from the correlation analysis according to the t test (two-sided) (b) of all possible combinations of postoperative parameters following multiple regression adjustment with the explanatory variables in Table 1 in the monofocal IOL group. The sample size for calculating the correlation coefficients is represented in (c).
(XLSX)

**S4 Table.** Pearson's correlation coefficients (a) and p values from the correlation analysis according to the t test (two-sided) (b) of all possible combinations of postoperative parameters following multiple regression adjustment with the explanatory variables in Table 1 in the rotationally asymmetric refractive IOL group. The sample size for calculating the correlation coefficients is represented in (c).
(XLSX)

**S1 Fig. Heatmap of Pearson's correlation coefficients between all possible combinations of variables in the ZCB00 group.** Pearson's correlation coefficients were adjusted by multiple regression with the explanatory variables in S1(b) Table. A significant correlation between two parameters is denoted by an asterisk (*) in this figure, with a Bonferroni-corrected $p < 0.00002$. To evaluate the significance of differences between two parameters, the two-sided t test was applied. S3(c) Table displays the sample size for each parameter. The illustration was created using R, a commercially available software program (version 3.6.1; R Core Team, 2019, Vienna, Austria) [22] (https://cran.r-project.org/web/packages/pheatmap/pheatmap.pdf).
(TIF)

**S2 Fig. Heatmap of Pearson's correlation coefficients between all possible combinations of variables in the LS-313 MF15 group.** Pearson's correlation coefficients were adjusted by multiple regression with the explanatory variables in S1(b) Table. A significant correlation between two parameters is denoted by an asterisk (*) in this figure, with a Bonferroni-corrected $p < 0.00002$. To evaluate the significance of differences between two parameters, the two-sided t test was applied. S4(c) Table displays the sample size for each parameter. The illustration was created using R, a commercially available software program (version 3.6.1; R Core Team, 2019, Vienna, Austria) [22] (https://cran.r-project.org/web/packages/pheatmap/pheatmap.pdf).
(TIF)

## Acknowledgments

We are truly grateful to all the staff at Tsukazaki Hospital for their participation in this study. The statistical outcomes and the study's design validity were rigorously evaluated by StaGen Co., Ltd., a professional data analysis company in Japan. They have provided a certificate confirming the review of the statistical methods by a professional statistician. This

manuscript's quality has been thoroughly verified by Research Square, which has issued a Methods and Data Reporting Certification. The language of the manuscript was edited, and the formatting was done by American Journal Experts (AJE), which has issued an editing certificate. Wiley Editing Services formatted the figures.

## Author contributions

**Conceptualization:** Hirotaka Tanabe, Tomofusa Yamauchi, Hitoshi Tabuchi.

**Data curation:** Hirotaka Tanabe, Tomohiro Shojo, Kosuke Takase.

**Formal analysis:** Hirotaka Tanabe.

**Investigation:** Hirotaka Tanabe.

**Methodology:** Hirotaka Tanabe.

**Project administration:** Hitoshi Tabuchi.

**Resources:** Hitoshi Tabuchi.

**Supervision:** Hirotaka Tanabe.

**Validation:** Hirotaka Tanabe.

**Visualization:** Hirotaka Tanabe.

**Writing – original draft:** Masaki Miyoshi, Hirotaka Tanabe.

**Writing – review & editing:** Hirotaka Tanabe.

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
