## [Decision Letter · Decision Letter 0]

11 Dec 2024

PONE-D-24-34677Comparison of visual performance between monofocal and rotationally asymmetric refractive intraocular lensesPLOS ONE

Dear Dr. Tanabe,

Thank you for submitting your manuscript to PLOS ONE. After careful consideration, we feel that it has merit but does not fully meet PLOS ONE’s publication criteria as it currently stands. Therefore, we invite you to submit a revised version of the manuscript that addresses the points raised during the review process.

Please check the quality of the figures as indicated by Reviewer 1 and consider the comments of Reviewer 2 in your revised version.

We look forward to receiving your revised manuscript.

Kind regards,

Timo Eppig

Academic Editor

PLOS ONE

Journal Requirements:

Reviewers' comments:

Reviewer's Responses to Questions

**Comments to the Author**

1. Is the manuscript technically sound, and do the data support the conclusions?

Reviewer #1: Partly

Reviewer #2: Yes

2. Has the statistical analysis been performed appropriately and rigorously? 

Reviewer #1: Yes

Reviewer #2: I Don't Know

3. Have the authors made all data underlying the findings in their manuscript fully available?

Reviewer #1: Yes

Reviewer #2: Yes

4. Is the manuscript presented in an intelligible fashion and written in standard English?

Reviewer #1: Yes

Reviewer #2: Yes

5. Review Comments to the Author

Reviewer #1: I suggest that improve the quality of Figures 1, 2 and 3 becuase it is somehow hard to reed the numbers and letters. However, based on my knowledge and understanding from the paper, I think the manuscript is prepared properly in the style of scientific paper.

Reviewer #2: Thank you for this paper, where you compare two lenses with an impressive number of subject.

Even though the number of patients is impressive, the findings are less surprising. Further I would rather look for papers, comparing specific monofocal IOLs with each other or multifocal IOLs with each other. Then one could really draw conclussions, that help patients and surgeons. Although the paper is well written and the statitics probably very good, I have difficulties recommending this paper for publications.

Please see my additional comments and questions below:

Abstract

Remove the list of higher order aberrations. It is sufficient to have it in the text.

Row 46/47:

rephrase “is designed to focus 95% of the time”. It is misleading.

Row 116/117:

“No boundary” is wrong. Specify clearly what is meant. Is there a sharp edge or a smooth transition zone?

Row 118:

“Twenty surgeons”

Did you look for different results according to the variable surgeon?

Row 134:

Why did you use more than 10 times the data that you calculate to be needed?

What is the benefit?

Statical analysis over time??? Were the first years as good as the last ones, were the better or worse years?

Table 1:

Please specify, whether it contains mean+-standard deviation or mean+-standard error with a confidence interval of … % in the caption

Please reduce the number of digits to the number of significant digits

Row 277-280:

How do you explain, that contrast sensitivity was superior in the Lentis group? One could assume that the monofocal wins this match.

Row 289:

Specify the meaning of a high Abbe number. What is the true effect of having an Abbe number of 55 or 57. Is there a real difference, that explains the contrast sensitivity? The eye itself already has a high chromatic aberration. I doubt, that the effect is really present.

6. PLOS authors have the option to publish the peer review history of their article (what does this mean? ). If published, this will include your full peer review and any attached files.

**Do you want your identity to be public for this peer review?** For information about this choice, including consent withdrawal, please see our Privacy Policy .

Reviewer #1: **Yes: ** Maziar Mirsalehi

Reviewer #2: No

---

## [Author Response · Author response to Decision Letter 1]

6 Mar 2025

Hirotaka Tanabe, MD, PhD

Department of Ophthalmology

Tsukazaki Hospital

68-1 Waku, Aboshi-ku, Himeji, Hyogo 671-1227, Japan

Tel.: +81-79-272-8555; Fax: +81-79-272-8550; Email: tennsyoudragon@icloud.com

Editor-in-Chief

March 6, 2025

Dear Editor-in-Chief:

We would like to attach our response to the editor and reviewers regarding our manuscript, “Comparison of visual performance between monofocal and rotationally asymmetric refractive intraocular lenses,” by Masaki Miyoshi, MD, Hirotaka Tanabe, MD, PhD, and colleagues, submitted as a research article for potential publication in Scientific Reports.

Editor comments

Dear Dr. Tanabe,

Thank you for submitting your manuscript to PLOS ONE. After careful consideration, we feel that it has merit but does not fully meet PLOS ONE’s publication criteria as it currently stands. Therefore, we invite you to submit a revised version of the manuscript that addresses the points raised during the review process.

Please check the quality of the figures as indicated by Reviewer 1 and consider the comments of Reviewer 2 in your revised version.

Response

Thank you very much for your excellent guidance. We have revised the manuscript in response to all the suggestions proposed in the review, and all revised text has been highlighted in yellow in a separate file. We genuinely appreciate your consideration of our manuscript.

Reviewer 1

I suggest that improve the quality of Figures 1, 2 and 3 becuase it is somehow hard to reed the numbers and letters. However, based on my knowledge and understanding from the paper, I think the manuscript is prepared properly in the style of scientific paper.

Response

Thank you for your positive comments and invaluable guidance for our manuscript. We have revised Figures 1-3 to improve legibility by removing the repetitions of the letters and redesigning them as much as possible while preserving the information. We have added the following sentences to the legends of Figures 1 and 3, respectively: "Non-WF (a), WF_4/6_post O (b), and WF_4/6_post I (c)." and "Nonvision/nondriving (a), vision (b), and driving (c)." We have also added the following sentence to the Acknowledgment section: "Wiley Editing Services formatted the figures.".

Reviewer 2

Thank you for this paper, where you compare two lenses with an impressive number of subject.

Even though the number of patients is impressive, the findings are less surprising. Further I would rather look for papers, comparing specific monofocal IOLs with each other or multifocal IOLs with each other. Then one could really draw conclussions, that help patients and surgeons. Although the paper is well written and the statitics probably very good, I have difficulties recommending this paper for publications.

Response

Thank you for your invaluable comments on our manuscript. In Japan, the ZCB00 monofocal IOL is fairly popular, while the Lentis Comfort IOL was the first nonmonofocal IOL covered by insurance; both have substantially impacted Japanese society. We have added the following sentences to the Introduction to address this issue: "In Japan, the ZCB00 is one of the most commonly used monofocal IOLs, whereas the LS-313 MF15, which features a rotationally asymmetric design and a +1.5 diopter near addition, was the first nonmonofocal IOL to be approved by insurance. Given their substantial impact on Japanese society, we conducted a retrospective study to compare the visual performance of these two types of IOLs."

Please see my additional comments and questions below:

Abstract

Remove the list of higher order aberrations. It is sufficient to have it in the text.

Response

Thank you for your invaluable guidance on our manuscript. We have removed the list of higher-order aberrations from the Abstract.

Row 46/47:

rephrase “is designed to focus 95% of the time”. It is misleading.

Response

Thank you for your invaluable guidance for our manuscript. We have revised the sentence to "The lens has an optical loss of only 5%, resulting in good postoperative contrast sensitivity."

Row 116/117:

“No boundary” is wrong. Specify clearly what is meant. Is there a sharp edge or a smooth transition zone?

Response

Thank you for your invaluable guidance for our manuscript. We have changed this text to "a smooth transition zone."

Row 118:

“Twenty surgeons”

Did you look for different results according to the variable surgeon?

Response

Thank you for your invaluable guidance for our manuscript. We confirmed that all surgeons at the hospital where this study was performed carefully followed the same protocols and that the procedures were thoroughly standardized. Therefore, we believe that the differences among surgeons were limited, and thus we did not add the surgeon as a random effect in the mixed-effects model employed in this analysis. We have added the following sentence to the Discussion section to address this issue: "We confirmed that all surgeons carefully adhered to the same protocols, ensuring thorough standardization of the procedures. As a result, we believe that variation among surgeons is minimal. Therefore, we did not include the surgeon as a random effect in the mixed-effects model used in this analysis when accounting for any potential biases in performance. The presence of multiple surgeons, each with a sufficient number of patients, resulted in a robust sample for our study."

Row 134:

Why did you use more than 10 times the data that you calculate to be needed?

What is the benefit?

Statical analysis over time??? Were the first years as good as the last ones, were the better or worse years?

Response

Thank you for your invaluable guidance for our manuscript. As you noted, 1,352 eyes from 676 patients met the selection criteria for this study, which is significantly larger than the required sample size determined from the power calculations on the basis of the logMAR effect size derived from previous studies. Since the objective of this study was also to conduct an exploratory analysis of trends for other observable factors, we chose to employ the entire population for our analysis. Another reason for this decision was to avoid any potential biases that could arise from selectively sampling the population under analysis. We have added the following sentence to the Discussion section to address this issue: "A total of 1,352 eyes from 676 patients met the selection criteria for this study, which is significantly larger than the required sample size determined from the power calculations on the basis of the logMAR effect size derived from previous studies. Since the objective of this study was also to conduct an exploratory analysis of trends for other observable factors, we chose to employ the entire population for our analysis. Another reason for this decision was to avoid any potential biases that could arise from selectively sampling the population under analysis."

Table 1:

Please specify, whether it contains mean+-standard deviation or mean+-standard error with a confidence interval of … % in the caption

Please reduce the number of digits to the number of significant digits

Response

Thank you for your invaluable guidance for our manuscript. Regarding the values in the table, we added a caption stating, "In this table, categorical variables are presented as the counts and frequencies, whereas continuous variables are presented with the number of samples (excluding those with missing data) and the mean ± standard deviation."

The number of significant digits for the mean and standard deviation has been modified to include one additional digit for the mean and two additional digits for the standard deviation.

Row 277-280:

How do you explain, that contrast sensitivity was superior in the Lentis group? One could assume that the monofocal wins this match.

Response

Thank you for your invaluable guidance for our manuscript. We have added the following sentence to the Discussion to address this issue: "In the present study, the contrast sensitivity of the rotationally asymmetric refractive IOL was comparable to or partially exceeded that of the monofocal IOL. The Lentis Comfort IOL is a multifocal intraocular lens designed to reduce optical loss. It features a smooth transition between the near and far portions of the lens, with only a mild difference in focus between each area; this design contributes to a mere 5% reduction in contrast sensitivity. Furthermore, unlike the ZCB00 IOL, the Lentis Comfort IOL is hydrophilic, which has been reported to increase the Abbe number [25]; indeed, although the Abbe number of the ZCB00 IOL is very high (55), that of the Lentis Comfort IOL is even greater (57) according to the manufacturer's reports. The Abbe number, or Vd-number, measures a transparent material's dispersion, indicating how the refractive index changes with wavelength. High Abbe numbers indicate low dispersion, which is crucial in the design of an IOLs; a lower Abbe number suggests greater chromatic aberration, resulting in decreased retinal image quality. The refractive indices of ocular media vary with wavelength, resulting in different focal points for colors and leading to chromatic aberrations and blurred images. The cornea and lens are the primary anatomical structures that contribute to these aberrations, with the crystalline lens responsible for approximately 28.5% of the total [26]. While typically manageable for patients with phakic eyes, longitudinal chromatic aberration can impact visual function in pseudophakic eyes equipped with low Abbe number IOLs, making the Abbe number of the IOL a key factor in contrast sensitivity differences [27,28]. For all the reasons listed above, the Lentis Comfort IOL seemed to have slightly but significantly better contrast sensitivity (6.3 degrees), contrast sensitivity with glare (4.0, 1.6, 1.0, 0.7 degrees), and significantly better contrast sensitivity with glare (6.3 degrees). In other words, there were no differences in most degrees of contrast sensitivity between the Lentis Comfort and ZCB00 IOLs in this study, despite the sufficient number of patients and rigorous statistical analysis involving a linear mixed-effects model with binocular data adjusted for age, sex, subjective refraction cylinder (CYL), subjective refraction spherical equivalent (SE), corneal astigmatism, axial length, pupil diameter, and corneal higher-order aberrations to ensure accurate results. Nevertheless, other confounding factors could explain the results."

References

25. Eppig T, Rawer A, Hoffmann P, Langenbucher A, Schröder S. On the chromatic dispersion of hydrophobic and hydrophilic intraocular lenses. Optom Vis Sci. 2020;97: 305-313.

26. Uosato HHirai HFukuhara JMatsuura T Fundamentals of Visual Optics [in Japanese]. Tokyo, Japan Kanahara & Co1990;

27. Negishi K, Ohnuma K, Hirayama N, Noda T. Effect of chromatic aberration on contrast sensitivity in pseudophakic eyes. Arch Ophthalmol. 2001 Aug;119(8):1154-8.

28. Zhao H, Mainster MA. The effect of chromatic dispersion on pseudophakic optical performance. Br J Ophthalmol. 2007 Sep;91(9):1225-9.

Row 289:

Specify the meaning of a high Abbe number. What is the true effect of having an Abbe number of 55 or 57. Is there a real difference, that explains the contrast sensitivity? The eye itself already has a high chromatic aberration. I doubt, that the effect is really present.

Response

Thank you for your invaluable guidance for our manuscript. We have added the following sentence to the Discussion to address this issue: "In the present study, the contrast sensitivity of the rotationally asymmetric refractive IOL was comparable to or partially exceeded that of the monofocal IOL. The Lentis Comfort IOL is a multifocal intraocular lens designed to reduce optical loss. It features a smooth transition between the near and far portions of the lens, with only a mild difference in focus between each area; this design contributes to a mere 5% reduction in contrast sensitivity. Furthermore, unlike the ZCB00 IOL, the Lentis Comfort IOL is hydrophilic, which has been reported to increase the Abbe number [25]; indeed, although the Abbe number of the ZCB00 IOL is very high (55), that of the Lentis Comfort IOL is even greater (57) according to the manufacturer's reports. The Abbe number, or Vd-number, measures a transparent material's dispersion, indicating how the refractive index changes with wavelength. High Abbe numbers indicate low dispersion, which is crucial in the design of an IOLs; a lower Abbe number suggests greater chromatic aberration, resulting in decreased retinal image quality. The refractive indices of ocular media vary with wavelength, resulting in different focal points for colors and leading to chromatic aberrations and blurred images. The cornea and lens are the primary anatomical structures that contribute to these aberrations, with the crystalline lens responsible for approximately 28.5% of the total [26]. While typically manageable for patients with phakic eyes, longitudinal chromatic aberration can impact visual function in pseudophakic eyes equipped with low Abbe number IOLs, making the Abbe number of the IOL a key factor in contrast sensitivity differences [27,28]. For all the reasons listed above, the Lentis Comfort IOL seemed to have slightly but significantly better contrast sensitivity (6.3 degrees), contrast sensitivity with glare (4.0, 1.6, 1.0, 0.7 degrees), and significantly better contrast sensitivity with glare (6.3 degrees). In other words, there were no differences in most degrees of contrast sensitivity between the Lentis Comfort and ZCB00 IOLs in this study, despite the sufficient number of patients and rigorous statistical analysis involving a linear mixed-effects model with binocular data adjusted for age, sex, subjective refraction cylinder (CYL), subjective refraction spherical equivalent (SE), corneal astigmatism, axial length, pupil diameter, and corneal higher-order aberrations to ensure accurate results. Nevertheless, other confounding factors could explain the results."

References

25. Eppig T, Rawer A, Hoffmann P, Langenbucher A, Schröder S. On the chromatic dispersion of hydrophobic and hydrophilic intraocular lenses. Optom Vis Sci. 2020;97: 305-313.

26. Uosato HHirai HFukuhara JMatsuura T Fundamentals of Visual Optics [in Japanese]. Tokyo, Japan Kanahara & Co1990;

27. Negishi K, Ohnuma K, Hirayama N, Noda T. Effect of chromatic aberration on contrast sensitivity in pseudophakic eyes. Arch Ophthalmol. 2001 Aug;119(8):1154-8.

28. Zhao H, Mainster MA. The effect of chromatic dispersion on pseudophakic optical performance. Br J Ophthalmol. 2007 Sep;91(9):1225-9.

Again, thank you very much for your consideration and excellent guidance for our manuscript. We look forward to hearing from you.

Sincerely yours,

Hirotaka Tanabe, MD, PhD

Chief Ophthalmologist (Cataract, Vitreous, Glaucoma, and Oculoplastic Surgeon)

Department of Ophthalmology, Tsukazaki Hospital

ORCID: https://orcid.org/0000-0002-1948-7408

---

## [Decision Letter · Decision Letter 1]

11 Apr 2025

Comparison of visual performance between monofocal and rotationally asymmetric refractive intraocular lenses

PONE-D-24-34677R1

Dear Dr. Tanabe,

We’re pleased to inform you that your manuscript has been judged scientifically suitable for publication and will be formally accepted for publication once it meets all outstanding technical requirements.

Kind regards,

Timo Eppig

Academic Editor

PLOS ONE

Additional Editor Comments (optional):

Reviewers' comments:

Reviewer's Responses to Questions

**Comments to the Author**

1. If the authors have adequately addressed your comments raised in a previous round of review and you feel that this manuscript is now acceptable for publication, you may indicate that here to bypass the “Comments to the Author” section, enter your conflict of interest statement in the “Confidential to Editor” section, and submit your "Accept" recommendation.

Reviewer #2: All comments have been addressed

2. Is the manuscript technically sound, and do the data support the conclusions?

Reviewer #2: Yes

3. Has the statistical analysis been performed appropriately and rigorously? 

Reviewer #2: Yes

4. Have the authors made all data underlying the findings in their manuscript fully available?

Reviewer #2: Yes

5. Is the manuscript presented in an intelligible fashion and written in standard English?

Reviewer #2: Yes

6. Review Comments to the Author

Reviewer #2: Thank you for your comments and the improved manuscript.

Except my last question, everything was addressed properly.

Anyway, from my point of view, the manuscript can be accepted.

7. PLOS authors have the option to publish the peer review history of their article (what does this mean? ). If published, this will include your full peer review and any attached files.

**Do you want your identity to be public for this peer review?** For information about this choice, including consent withdrawal, please see our Privacy Policy .

Reviewer #2: No

---

## [Editor Report · Acceptance letter]

PONE-D-24-34677R1

PLOS ONE

Dear Dr. Tanabe,

I'm pleased to inform you that your manuscript has been deemed suitable for publication in PLOS ONE. Congratulations! Your manuscript is now being handed over to our production team.

Kind regards,

on behalf of

Prof. Dr. Timo Eppig

Academic Editor

PLOS ONE